# Gelation Impairs Phase Separation and Small Molecule Migration in Polymer Mixtures

**DOI:** 10.3390/polym12071576

**Published:** 2020-07-16

**Authors:** Biswaroop Mukherjee, Buddhapriya Chakrabarti

**Affiliations:** Department of Physics and Astronomy, University of Sheffield, Sheffield S3 7RH, UK; b.mukherjee@sheffield.ac.uk

**Keywords:** surface migration, polymer gels, wetting phenomena, mesoscale simulations, polymer theory

## Abstract

Surface segregation of the low molecular weight component of a polymeric mixture is a ubiquitous phenomenon that leads to degradation of industrial formulations. We report a simultaneous phase separation and surface migration phenomena in oligomer–polymer (OP) and oligomer–gel (OG) systems following a temperature quench that induces demixing of components. We compute equilibrium and time varying migrant (oligomer) density profiles and wetting layer thickness in these systems using coarse grained molecular dynamics (CGMD) and mesoscale hydrodynamics (MH) simulations. Such multiscale methods quantitatively describe the phenomena over a wide range of length and time scales. We show that surface migration in gel–oligomer systems is significantly reduced on account of network elasticity. Furthermore, the phase separation processes are significantly slowed in gels leading to the modification of the well known Lifshitz–Slyozov–Wagner (LSW) law $\ell \left(\tau \right)∼\tau^{1/3}$. Our work allows for rational design of polymer/gel–oligomer mixtures with predictable surface segregation characteristics that can be compared against experiments.

## 1. Introduction

Complex mixtures of soft materials, used in industrial formulations, are often plagued by the migration of the small molecular weight component to the interface open to atmosphere [1]. Such “surface segregation” of the active ingredients in a formulation leads to loss of function in a variety of our daily products [2,3] e.g., adhesive loss in feminine and hygiene care products, flaking behaviour of paints, and blooming of fat in chocolate. The basic phenomenology of surface segregation/wetting can be understood in a model binary polymer mixture of different molecular weights and a surface exposed to atmosphere. The surface composition of a mixture is determined by the balance between their relative surface energies and the the inter-facial energy between the two phases [4]. A loss of entropy and gain in surface energy of a molecule dictates the equilibrium surface fraction.

For well-mixed systems having a free surface, the segregation profile with oligomer concentration monotonically decreasing as a function of depth (≈exp(−z/ξ)) is observed. In contrast, a macroscopic wetting layer forms, with wetting layer thickness $\ell_{w}\approx$ 100–200 nm, for systems for which the bulk thermodynamic phase is de-mixed [5,6,7]. The classic experiments demonstrating surface directed spinodal decomposition (SDSD) were performed on an unstable polymer mixture of PEP and dPEP having a free surface, which preferentially attracts dPEP [8,9,10]. While these experiments have been performed for 50:50 polymeric mixtures, where both polymers have similar lengths, we are interested in studying migration kinetics in asymmetric mixtures (both in terms of chain length and composition), where the oligomer concentration is small. Mean field (MFT) [11] and self-consistent field theories (SCFT) that combine the bulk thermodynamics of polymer mixtures with that of a surface expressed in terms of phenomenological free energy functionals have been employed to compute the surface migrant fraction and wetting layer thickness for well mixed and de-mixed systems with moderate success. These thermodynamic theories, however, do not describe how the migrant concentration profiles and wetting layers evolve as a function of time [12,13].

The kinetics of surface-directed spinodal decomposition (SDSD) is a rich non-equilibrium, many-body phenomena where the dynamic effects of surface wetting and bulk phase separation are coupled in a non-trivial fashion [14,15,16,17,18,19]. An early time surface interaction specific growth law that leads to late time LSW kinetics is observed for the minority component being attracted by the surface [19]. Experiments for SDSD for OP systems show that the wetting layer thickness grows logarithmically as a function of time for a shallow quench and follows [20,21] LSW growth for a deep quench. Several factors, e.g., surface adsorption [22,23], surface roughness [24,25], and confinement, can modify surface migration kinetics in polymer mixtures leading to novel phenomena, e.g., lateral phase separation [25,26].

Within the biological milieu, the phase separation of proteins within living cells is a very active area of research [27,28]. While the physics of droplet growth is well studied when the surrounding matrix is a simple liquid, relatively little is known about droplet growth kinetics in an elastic environment like the cell cytoplasm. This is an exciting recent area with lots of experimental activity [29,30] but relatively scant theoretical understanding. In an earlier theoretical study [31], we showed that increasing the bulk modulus of a gel–oligomer mixture causes a dramatic reduction in the surface fraction of migrant molecules. The wetting transition observed for de-mixed systems can also be avoided. This study was based on a mean field analysis of a phenomenological free energy functional. We augment this study with CGMD simulations which gives a more accurate representation of the physical situation particularly near a phase transition.

In this paper, we report the kinetics of surface migration of small molecules (oligomers) in an (*a*) oligomer–polymer (OP) and (*b*) oligomer–gel (OG) mixture undergoing phase separation following an instantaneous temperature quench that renders the mixed phase unstable. The surface free energy difference preferentially attracts oligomers. We (i) compute dynamic surface concentration profiles of oligomers, (ii) quantify the difference in bulk coarsening phenomena as a function of depth of quench ∆T and gel bulk modulus B, and (iii) conclusively demonstrate that surface migration of oligomers in an end-linked polymer gel is suppressed in comparison to a polymer–oligomer mixture using (i) coarse grained molecular dynamics (CGMD) and (ii) mesoscale hydrodynamics (CHC) simulations (see Appendix A for details).

## 2. Materials and Methods

We perform CGMD simulations of 10:90 (i.e., 10% oligomer) OP and OG systems using a Kremer–Grest type bead spring model [32] using GROMACS [33]. The gel matrix of the OG system is constructed by permanently cross-linking terminal beads of two polymers that lie within a distance $\zeta \approx R_{0}$, where $R_{0}$ is the bond distance, (see Appendix A for details). The mesh size of such a system is tuned by changing the relative volume fraction of polymers that make up the network. Interaction strengths among *A* and *B* polymers are chosen such that $\epsilon_{AA}=\epsilon_{BB}=2\epsilon_{AB}=\epsilon$. This choice of energy-scales ensure that the mixture spontaneously phase separate upon a quench from the initial high temperature to its final low temperature configuration. The system is equilibrated in a box following a temperature quench with periodic boundary conditions along *x* and *y* directions and two walls placed at z=0 and z=d. The wall at z=0 preferentially attracts the oligomers (modeling differing surface free energies of oligomers) while the wall at z=d is neutral to both species (see Appendix A).

Configuration snapshots of OG system undergoing simultaneous phase separation and surface migration, obtained from CGMD simulations at different scales of resolution, are shown in Figure 1. The system is quenched from a high temperature $T_{i}=10$ to $T_{f}=1$ (in dimensionless units) and evolved for $\tau =\tau_{LJ}\times 10^{5}$ to ensure thermodynamic equilibrium. Compared to the OP system (see Appendix A), the phase-separation and thereby surface migration process in arrested in gels. This is evidenced by the presence of (*a*) more oligomer droplets that are smaller in size in comparison to OP systems, (*b*) thinner wetting layer, and (*c*) a narrower depletion region just below the wetting layer (see final configuration of oligomers in Appendix A). The arrested coarsening and migration behavior is seen in panels (b) and (c) where the oligomer droplets are stuck in a cage formed by end linked polymers.

Since phase separation is inherently a “slow” phenomena, it is difficult to faithfully model it for experimental time scales using detailed CGMD simulations. Meso-scale simulations, however, access much larger length-scales and longer time-scales. We therefore augment our CGMD simulations with a mesoscale model of phase separation dynamics with the Flory–Huggins free energy functional describing the bulk thermodynamics and local potentials mimicking the preferential surface affinity of oligomers. As the oligomers do not evaporate out of the system, the number of polymers and oligomers in our system is conserved. We therefore employ a time-dependent Ginzburg–Landau formalism using model *B* dynamics that takes into account the conserved nature of the order parameter [34,35]. We solve the nonlinear diffusion equation for the order-parameter field, i.e., dynamic oligomer concentration profiles with appropriate boundary conditions to match against similar data obtained from CGMD simulations.

The dynamic concentration profiles of oligomers ϕ(r,t) for a polymer–oligomer and gel–oligomer system satisfies
(1)$$\frac{\partial \phi (r,t)}{\partial t}=\nabla \cdot \left[M\nabla \frac{\delta F[\phi (r,t)]}{\delta \phi (r,t)}+\theta \left(r,t\right)\right],$$
where *M* is the mobility, assumed to be composition independent and the local chemical potential $\mu \left(\phi \left(r,t\right)\right)=\frac{\delta F[\phi (r,t)]}{\delta \phi (r,t)}$. An additive vectorial conserved noise θ(r,t) in Equation (Equation 1) modelling solvent effects, satisfying $\langle \theta_{i}\left(r,t\right)\rangle$ = 0, and $\langle \theta_{i}\left(r,t\right)\theta_{j}\left(r^{\prime },t^{\prime }\right)\rangle =2Mk_{B}T\delta_{ij}\delta \left(r-r^{\prime }\right)\delta \left(t-t^{\prime }\right)$ ensures thermodynamic equilibrium at long times. Since the average concentration of polymers/gel and oligomers in our system is far from the symmetry point $\phi_{\infty }=1/2$ and in this regime domain coarsening occurs primarily via diffusion, we do not include explicit hydrodynamics interactions [36] in our meso-scale model.

The free energy functional for an in-compressible binary fluid mixture, in two space dimensions, confined between selectively attracting walls (surfaces), located at z=0 and z=d is given by
(2)$$\begin{array}{ccc} F\left[\phi \left(r\right)\right]/k_{B}T=\frac{1}{a^{2}}\int_{0}^{d}\int_{0}^{d}[f_{FH}\left(\phi \right) & + & k\left(\phi \right){\left(\nabla \phi \right)}^{2}+f_{0}\left(\phi \right)\delta \left(z\right)\\ & + & f_{d}\left(\phi \right)\delta \left(z-d\right)]dxdz, \end{array}$$
where *F* is the free-energy, and *z* and *x* are the coordinates perpendicular and parallel to the wall, respectively, and *a* is the Flory–Huggins lattice spacing. The first term in Equation (Equation 2) is the bulk free energy and the second term accounts for energy costs associated with the spatial gradients of the composition field with a stiffness coefficient $k\left(\phi \right)=\frac{a^{2}}{36\phi (1-\phi )}$. Note that surface free-energies $f_{0}\left(\phi_{1}\right)$ and $f_{d}\left(\phi_{D}\right)$ have dimensions of length such that F[ϕ(r)] is dimensionless. The functional forms of $f_{0}\left(\phi_{1}\right)=h_{0}\phi_{1}+\frac{1}{2}g_{0}\phi_{1}^{2}$ and $f_{d}\left(\phi_{D}\right)=h_{D}\phi_{D}+\frac{1}{2}g_{D}\phi_{D}^{2}$, respectively. As in our CGMD simulations, the wall at z=0 attracts the oligomer *B* while the wall at z=d is neutral to both the components. We study the approach to equilibrium, when this mixture is quenched to the two phase region, starting from an initial uniform phase, which is thermodynamically unstable, for a OP and OG mixture, with the component *A* (having local composition ϕ(r,t)) denoting the polymer with degree of polymerisation, $N_{A}$ or the gel, and an oligomer *B* (with local composition (1−ϕ(r,t))), with degree of polymerisation, $N_{B}$, respectively.

The dimensionless Flory–Huggins free energy for a polymer–oligomer mixture is given by, Equation (Equation 3),
(3)$$f_{FH}\left(\phi \right)=\frac{\phi }{N_{A}}\ln\left(\phi \right)+\frac{\left(1-\phi \right)}{N_{B}}\ln\left(1-\phi \right)+\chi \phi \left(1-\phi \right),$$
and the Flory–Rehner free energy describing the gel–oligomer mixture is given by, Equation (Equation 4),
(4)$$\begin{array}{ccc} f_{FHE}\left(\phi \right) & = & \frac{\left(1-\phi \right)}{N_{B}}\ln\left(1-\phi \right)+\chi \phi \left(1-\phi \right)\\ & + & B\left(\frac{\phi_{\infty }}{2}\right)\left[{\left(\frac{\phi }{\phi_{\infty }}\right)}^{2/3}+2{\left(\frac{\phi }{\phi_{\infty }}\right)}^{1/3}-3\right], \end{array}$$
where χ is the Flory–Huggins interaction parameter. The bulk concentration of the polymers that make up the gel is denoted by $\phi_{\infty }$, which is identified here as the region in the vicinity of z=d, and *B* denotes the bulk modulus of the gel. The precise connection between the bulk modulus and the microscopic gel architecture is not known, to the best of our knowledge. Therefore, we use values of the bulk modulus which are similar to those used in earlier calculations on the thermodynamics of phase separation in mixtures of small molecules and gels [31]. In an ongoing work, we are investigating the effects of gel elasticity on the domain coarsening length scale and how this depends on the gel fraction or the cross-link density of the mesh structure. We set the value of the Flory–Huggins χ parameter to 1.1$\chi_{sp}$, where $\chi_{sp}$ is its value at the spinodal. This choice makes the initial uniform state unstable and the system evolves to its new phase-separated equilibrium state in the presence of the external surface that prefers one of the components. We numerically integrate Equation (Equation 1) for both forms of the free energies Equations (Equation 3) and (Equation 4) with the boundary conditions described earlier (see Appendix A for details). For bulk simulations, we impose a periodic boundary condition along all directions, while in the presence of walls we impose a zero flux boundary condition at the walls and periodic boundary condition along the lateral dimensions. This ensures that the order parameter is conserved throughout the evolution process. The initial ϕ(r,t) field configuration for meso-scale simulations on a (L×L) lattice with L=50 is chosen to be ϕ(r,0) = $\phi_{\infty }$ + δϕ, with $\phi_{\infty }$ being the initial concentration of a polymer/gel and δϕ is a uniformly distributed random number in the interval $\left[-0.05,0.05\right]$. $\phi_{\infty }$ is set to 0.7, which signifies a 30:70 mixture of oligomer–polymer and oligomer–gel. $N_{A}$ and $N_{B}$ are chosen as 100 and 50, respectively.

We coarse-grain particulate configuration snapshots of CGMD simulations [37] to obtain oligomer concentration ϕ(r,t) to compare against MH results following a majority rule. The simulation box is divided into cubes of size σ≈b, where *b* is the bead diameter and the number of monomers belonging to polymer $n_{A}$ and oligomer $n_{B}$ counted. The coarse-grained order parameter field for the *i*-th cell $\phi_{i}=+1$ if $n_{A}>n_{B}$, otherwise $\phi_{i}=-1$. For the simulations in the presence of walls, the coarse-grained $\phi_{i}$’s, the one-dimensional density of oligomers, as a function of the depth from the upper wall, is obtained by performing an average over the two lateral dimensions. The equal time spatial correlation function in bulk mixtures
(5)$$\begin{array}{c} C(r,\tau )=\langle \phi (0,\tau )\phi (r,\tau )\rangle -\langle \phi (0,\tau )\rangle \langle \phi (r,\tau )\rangle , \end{array}$$
where *r* is the radial distance between the centres of two cubes, and τ is the time elapsed since the instantaneous quench, is used to study the phase-separation dynamics. The angular brackets in Equation (Equation 5) indicate averaging over independent initial configurations and the first zero crossing of C(r,τ) determines the domain size ℓ(τ).

## 3. Results

The time-dependence of the coarsening length-scale is shown in Figure 2, with (a) and (b) showing results from bulk MD simulations and bulk mesoscale simulations, respectively, with filled circles denoting coarsening in polymer–oligomer mixture and filled squared denoting coarsening in a gel–oligomer mixture with bulk modulus, B=0.05. In both cases, the domain size initially grows as a function of time as $\ell \left(\tau \right)∼\tau^{1/3}$ following a Lifshitz–Slyozov law (see Materials and Methods for a definition of ℓ(τ)). At longer times, the phase separation process is arrested in gels showing ℓ(τ) saturating as a function of τ (see the blue squares in panels (a) and (b) of Figure 2). The saturation value and the time at which ℓ(τ) saturates depend on the elasticity of the gel-matrix. Recent experiments on arrested droplet growth in the presence of an elastic matrix show how the saturation size of the droplets monotonically increases as the matrix becomes softer [30].

In order to match our results with CGMD simulations, we also perform meso-scale simulations, where we simulate the phase separation kinetics in a 30:70 asymmetric mixture of oligomers and polymers/gel having polymerisation index $N_{A}=100$, and $N_{B}=50$, respectively (see Materials and Methods for a description of the system). For this composition, the parameter $\phi_{\infty }$ is set to 0.7 in the gel–oligomer free-energy in Equation (Equation 4). We set $\chi =1.1\chi_{sp}$, where $\chi_{sp}$ corresponds to the value of the Flory parameter at the spinodal, for the above parameters and $\phi_{\infty }=0.7$. Two values of bulk modulus B=0.05 and B=0.1 have been used and the computed thermodynamic quantities are averaged over $N_{r}=10$ different initial configurations. We numerically integrate the non-dimensionalised version of Equation (Equation 1) accounting for the conserved noise following a forward Euler scheme (see Appendix A).

In presence of a top surface at z=0, which preferentially attracts the oligomers a complete wetting transition is observed for the OP system while partial wetting is observed for the OG systems. The migrant concentration configurations, in the meso-scale description, close to equilibrium having a small chemical potential gradient δμ≈0, obtained by numerically integrating Equation (Equation 1) for long times t→∞ for both systems are shown in Figure 3. At long times, the phase separation is nearly complete for OP systems resulting in the formation of a thick wetting layer. In contrast, the coarsening process is arrested in gels resulting in a diffuse thin wetting layer that decreases monotonically on increasing the bulk modulus, (c) and (d). These results can be understood from the variation of the bulk free energy as a function of the oligomer concentration ϕ for both systems. In the absence of elastic interactions, the system admits two minima, with well separated ϕ values, corresponding to an equilibrium phases that are A/B rich. For a gel, elastic interactions result in lowering the free energy of an oligomer rich state. If the surface affinity of the oligomers (set by $g_{0}$, $h_{0}$, $g_{D}$ and $h_{D}$ (see Materials and Methods for a definition of these quantities)) is insufficient to overcome the cost of elastically deforming a polymeric cage that traps the oligomer droplets, the equilibrium state is one with oligomers inside the gel resulting in a thinner and diffuse wetting layer.

Figure 4 shows the time evolution of oligomer concentration with an attractive surface. Panels (a) and (b), respectively, show profiles obtained from MD simulations for a gel-fraction of 0.9, following an instantaneous quench from an initial temperature $T_{i}=10$ to $T_{f}=1$. The migrant density in the vicinity of the upper wall at z=0 is ϕ(z)>0.5 for the OP mixture in panel (a), whereas, ϕ(z)<0.5 for the OG system in panel (b). Similarly, (c) and (d) of Figure 4 show density profiles obtained from mesoscale simulations for parameters ($\phi_{\infty }=0.7$, and $\chi_{sim}=1.1\chi_{sp}$) as described above. The characteristic time and space discretisation scales of the mesoscale simulations are dependent on the thermodynamic state point (see Appendix A). When comparing results from different simulations, we have rescaled the raw data such that all temporal and spatial scales in Figure 4 are equal. These profiles clearly show that the surface migration is indeed significantly suppressed due to the gelation, both in the CGMD description (compare profiles in panels (a) and (b) of Figure 4) and the meso-scale description (compare profiles in panels (c) and (d) of Figure 4). Experiments on the SDSD in mixtures of poly(ethylene-propylene)(PEP) and per-deuterated poly(ethylene-propylene) (d-PEP) [8] show surface segregation profiles which are very similar to the profiles obtained via our simulations.

The oligomer density profiles shown in Figure 4 are used to compute the migrant fraction at the attractive surface as a function of time. In CGMD simulations, this is computed by counting the number of particles between z=0 and the first minimum of the density profile ϕ(r,t) at z≈4σ. In meso-scale simulations, the measure of the migrant fraction is provided by the expression: $\phi_{1}=\int_{0}^{\ell_{w}}\left(\phi (z)-0.5\right)dz$, i.e., the area of ϕ(z) above the line ϕ(z)=0.5 and $\ell_{w}$ is the thickness of the wetting layer in the meso-scale simulations, which is defined as the distance from the surface when ϕ(z) falls below 0.5 in (c) and (d) of Figure 4. Figure 5a,b shows the time variation of the migrant fraction for the OP and OG systems by the CGMD and the meso-scale simulations, respectively. These results corroborate the results shown in Figure 4 and panel (a) clearly demonstrates that the migrant fraction at the longest simulated time decreases almost by a factor of two for the OG, relative to the OP system in our CGMD simulations. In panel (b), the results from meso-scale simulation confirm this observation and we further observe that the migrant fraction at the longest time reduces as one increases the stiffness of the gel, which is quantified by the value of the bulk modulus. These results are very similar to those observed in the recent experiments on droplet growth within an elastic matrix whose elastic modulus is tuned to control the droplet size [30]. As evidenced in the oligomer density profiles, increasing the bulk modulus causes a decrease in the migrant fraction. A similar dramatic slow growth of the wetting layer thickness as a function of increasing bulk modulus is observed in CGMD simulations (see Appendix A).

## 4. Conclusions

In conclusion, we have showed that surface migration in OG systems can be significantly reduced by increasing the elastic modulus of the gel. Our CGMD simulations show that for these systems oligomer droplets are stuck in the gel meshwork, leading to a phase-separation arrest that modifies both the domain growth law and surface segregation kinetics. The phase separation proceeds either via nucleation, i.e., growth and coalescence of droplets or spinodal decomposition, i.e., unstable growth. The nucleation and growth phenomena are driven by surface tension and mass diffusion among different sized droplets and leads to the LSW domain growth law $\ell \left(\tau \right)∼\tau^{1/3}$ for systems with Ising symmetry as seen from simulations [38] and asymptotic analysis [39,40]. Such a growth law has been explored for binary [41] and multi-component fluids [37] and polymer mixtures [42,43,44,45,46,47], and has been verified in experiments [48,49] (see [38] for a review). End linking polymers result in effectively slowing down relaxation mechanisms leading to a “dynamical asymmetry” among the constituents leading to a dramatic slow down of the LSW law. The system thus exhibits characteristics of viscoelastic phase separation [36,50]. While size disparity in soft material mixtures leading to formation of transient networks [51] is common, it is fundamentally different from our case where the network structure is permanent. Departures from the LSW kinetics in bulk fluids can also occur in systems where phase separation is coupled to chemical reactions. A coupled gelation, phase-separation, and surface migration study with competing time-scales that lead to novel phenomena will be reported in a future study.

The generic multi-scale framework developed in this paper is suitable for rational design of oligomer–polymer/gel mixtures with predictable surface migration and phase behaviour. Our non-equilibrium phase ordering kinetics presented here can also be extended to the biological domain via the incorporation of an active term. In particular, we believe that the techniques and results presented here are directly applicable to viscous cellular environments where membrane-less organelles sort cellular components via liquid–liquid phase separation [52,53].

## Figures and Tables

**Figure 1 polymers-12-01576-f001:**
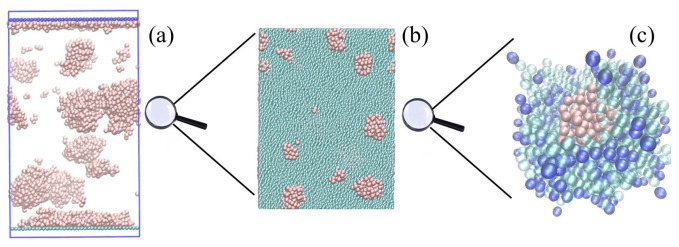
Configuration snapshots of a gel–oligomer system (CGMD) undergoing phase separation at different scales of resolution. (**a**) shows the simulation box with oligomers droplets (pink), while (**b**) shows a magnified region of oligomer droplets (pink) in a gel-matrix (green); (**c**) shows an oligomer droplet (pink beads) trapped within a mesh of an end-linked gel formed by polymers (green beads) with permanently stuck end groups (blue beads).

**Figure 2 polymers-12-01576-f002:**
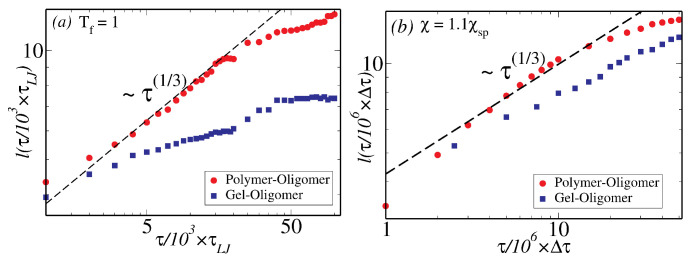
The time-dependence of the coarsening length as a function of time computed from bulk simulations for polymer–oligomer mixture (shown in red filled circles) and gel–oligomer mixtures (shown in blue-filled squares) simulated via MD simulations (**a**)) and via meso-scale simulations, where the gel has a bulk modulus, B=0.05 (**b**).

**Figure 3 polymers-12-01576-f003:**
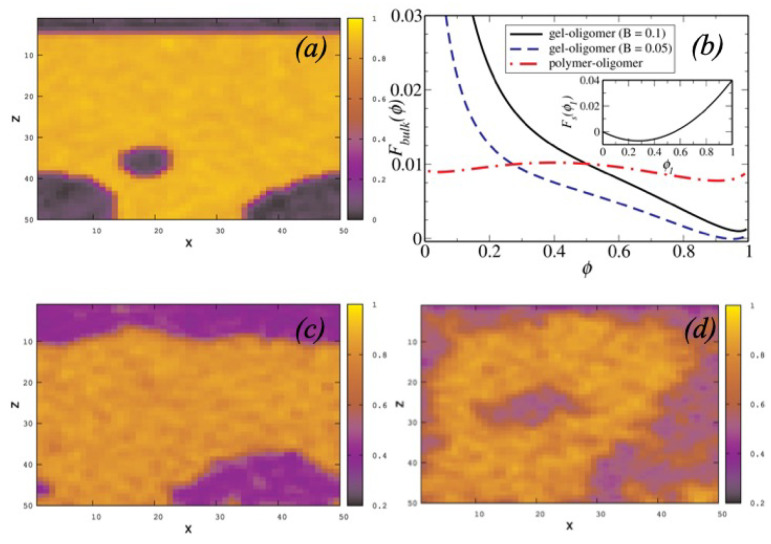
Oligomer concentration configurations for OP (**a**) and OG (**c**,**d**) systems showing polymer rich (yellow) and oligomer rich (dark) domains obtained from mesoscale simulations with oligomers wetting the wall at z=0. (**b**) shows Flory–Huggins (red dashes) and Flory–Rehner (blue dash-dotted line B=0.05, and black solid line B=0.1) forms of bulk free energies corresponding to these mixtures; (**b**) shows variation of surface free energy of oligomers for interaction parameters $g_{0}=0.18$ and $h_{0}=-0.05$ as a function of the surface oligomer concentration $\phi_{1}$.

**Figure 4 polymers-12-01576-f004:**
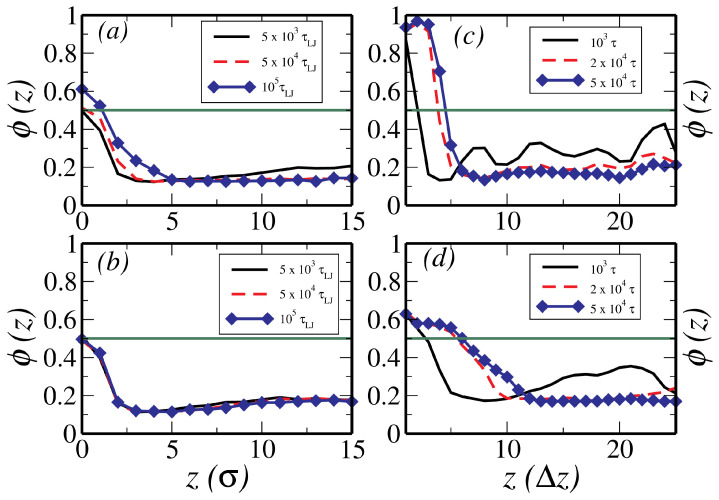
The time-evolution of the oligomer density computed from MD simulation in a polymer–oligomer system (**a**) and a gel–oligomer system (**b**) following a quench from an initial temperature of $T_{i}=10$ to a final temperature of $T_{f}=1$; (**c**,**d**) show the time evolution of the oligomer density computed from meso-scale simulations, following a quench to the two-phase region, for a polymer–oligomer and a gel–oligomer mixtures, respectively.

**Figure 5 polymers-12-01576-f005:**
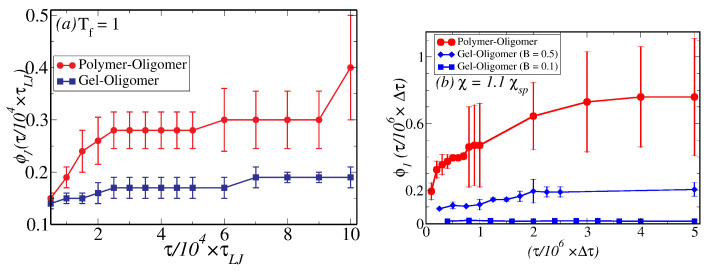
The fraction of migrant molecule computed from an MD simulation of a polymer–oligomer mixture and a gel–oligomer mixture is shown in (**a**), while (**b**) shows the same from a meso-scale mixture of a polymers and oligomers and a mixture of gel and oligomers for two values of gel bulk moduli.

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
