# Peer review of "Gelation Impairs Phase Separation and Small Molecule Migration in Polymer Mixtures"

_polymers, 2020, doi:10.3390/polym12071576_

Round 1
Reviewer 1 Report
The authors report a simultaneous phase separation and surface migration phenomena in oligomer-polymer (OP) and oligomer-gel (OG) systems. Several parameters are calculated in these polymer mixture systems using the multi-scale computer simulation methods. The results show that surface migration in gel-oligomer systems is significantly reduced on account of network elasticity. The results are meaningful and interesting. I recommend publication of the manuscript only after the minor revisions: (1) some newly references, about the migration, polymer mixtures, should be cited in the manuscript; (2) the comparisons between their results and previous experiment works should be discussed in more detailed in the manuscript.
Author Response
We thank the referee for his encouraging comments on the manuscript.
We are not aware of any experimental work monitoring small molecule surface migration behaviour in polymer-gels. However, experiments describing the interplay of elastic effects and phase separation of biopolymer complexes e.g. proteins in cellular environments have been carried out. The results from these studies are relevant in understanding biological organisation via membrane-less organelles inside cells. We have added a number of these references in the Introduction (References 12 to 15) and Conclusion section of the modified manuscript. These changes have been highlighted in blue.
With regards to the correspondence between our theoretical work and experiments, we would like to point out the similarities between the dynamic oligomer profiles, in polymer-oligomer mixtures, obtained from our multi-scale simulations (CGMD and mesoscale simulations) and an earlier experimental paper (Reference 8 of the manuscript, Jones et al. Phys. Rev. Lett.1991, 66, 1326–1329). We present the experimental data on the left panel and our results on the right panel for comparison in the picture below. This discussion has been added in the Results section of the modified manuscript and the text has been highlighted in blue.
Figure 1. Left panel shows volume fraction vs depth profiles of d-PEP in an initially uniform mixture with a d-PEP volume fraction of 0.5 after aging at 35 C for 19200 s. This is compared against both meso-scale and CGMD simulations (Fig. 4) of our manuscript.

Reviewer 2 Report
All of the simulations and calculations presented in the manuscript seem to be coherently planned and executed. However, the Authors treat all polymers a single material with the same molecular weight. It is impossible to apply one calculation uniformly and indiscriminately in order to study materials that differ in structure and properties.
Moreover, calculations that are not supported by results obtained from tangible samples have very little significant value.
For this reason, I suggest a rejection of the submitted manuscript.
Author Response
Response:
We thank the referee for his critical reviewing of the manuscript. The referee questions the applicability of our study stating that we have chosen polymers with the same molecular weight in our simulations.
We agree that real polymeric mixtures will have some degree of polydispersity which will affect the phase separation. Phase separations in polydisperse mixtures is well studied (see Cates, Warren, Clarke, etc.). However, the referee does not put forward any argument to substantiate his belief that polydispersity will change the basic result, i.e. arresting phase separation and surface migration in a gel.
So far we can see, the modification for a polydisperse sample is a change in the average mesh size of the gel, which will modify its bulk modulus. Therefore a higher fraction of cross-linkers might be required to have the same reduction in migration as a monodisperse system studied here. There is no reason to believe that polydispersity will qualitatively change the results presented in the manuscript.
We agree with the referee's comments that "calculations that are not supported by results obtained from tangible samples have very little significant value."
This is indeed why we believe that our paper should be published, namely, to prompt the experimentalists to test our predictions, even with polydisperse samples.
It is worth noting in this context that though our CGMD polymer-oligomer simulations do not possess any polydispersity, the gel structure is quite inhomogeneous as the gel has been prepared from an initially homogeneous polymer-oligomer mixture at high temperature, whose terminal beads have been bonded if they satisfy certain geometrical criteria, discussed in the text.
This procedure automatically introduces a distribution of mesh sizes in the simulated gel structure. Additionally we have performed an ensemble of quenches from several independent starting configurations and reported properties have been averaged over trajectories from these various initial configurations. There is heterogeneity in these initial configurations and as a result the exact number of cross-links vary for each independent quench. Thus we think that heterogeneity effects associated with polydispersity has already been accounted for, to a certain extent, in our CGMD simulations.
Reviewer 3 Report
The article by B. Mukherjee and B. Chakrabarti is devoted to investigation of surface segregation of some low molecular weight component in polymeric mixtures. The authors report results, which have potential interest for rational design of polymer/gel-oligomer mixtures and may help to improve surface characteristics of polymer coatings.
This article may be published after the major revision. The most significant drawback of the work is its structure. The authors present the results immediately after the introduction. With such a construction of the article, it is very difficult to understand what and how was done, since a reader need to refer to sections that have not yet been read. As a result, one have to repeatedly navigate through the text, which makes reading the article very difficult. In my opinion, it is better to use the following manuscript organization would be used: Introduction, Materials and Methods, Results and Discussion.
Less significant comments. The choice of model parameters for bulk modulus (B), chi parameters and interaction strength among A / B polymers should be thoroughly described.
Author Response
We thank the referee for his review.
We had modified the structure of our paper to suit the style of the journal. We are resubmitting the paper in accordance with the referee’s suggestion and look forward to an editorial approval of our manuscript.
To address the technical points raised by the referee we note that a precise connection between the bulk modulus and the microscopic gel architecture is not known. Hence we use values of the bulk modulus which are similar to those used in earlier calculations on the thermodynamics of phase separation in mixtures of small molecules and gels. In an ongoing work, we are investigating the effects of gel elasticity on the domain coarsening length scale and how this depends on the gel fraction or the cross-link density of the mesh structure. This discussion is highlighted in blue and appears on Page 7 of the modified manuscript.
In the mesoscale simulations, the parameter has been chosen to be 10% higher than a value at which spinodal decomposition occurs. This makes the initial uniform state unstable and the system evolves to its new phase-separated equilibrium state in the presence of the external surface that prefers one of the components. Similarly, in the CGMD simulations the epsilon parameters have been chosen so that the system is deemed to phase separate upon an instantaneous quench from an initially high temperature to the final low temperature state. A discussion regarding these issues have been added to the modified manuscript in Materials and Methods section.
Reviewer 4 Report
Using a combination of coarse grained molecular dynamics (CGMD) and mesoscale hydrodynamics (MH) simulations Mukherjee and Chakraborty in this paper present a comparative study on the phase separation and surface migration in oligomer-polymer (OP) and oligomer-gel (OG) systems. Both the phase separation and surface migration is found to be reduced in the OG system. The study is of interest and the paper written in a lucid way. I therefore recommend accepting this work for publication essentially in its current form. I just have a minor suggestion for the authors about the title. While the authors study the important phenomena of surface migration and phase separation in the work, the title seems to leave out the latter mentioning the effect of gelation only on the migration. Perhaps the authors can chose a more exclusive title.
Author Response
Response:
We thank the referee for his positive views on our manuscript. We have changed the title of the manuscript to “Gelation impairs phase separation and surface migration in complex mixtures.”
Reviewer 5 Report
The present manuscript deals with molecular simulation about migration phenomena in oligomer-polymer and oligomer-gel systems. Some points should be addressed.
Please add a reference in line 24 and line 36
Please define lw and all abbreviation at the first appearance in the manuscript
The results presented in figure 1-5 must be detailed. They are very concise.
Why standard deviation is reported only for points in figure 5A? Please consider if reporting standard deviation also in the other plots.
I am not convinced that it is possible to affirm that "surface migration in OG systems can be significantly reduced by increasing the elastic modulus of the gel" on the base of the simulation performed in the study. Please discuss more the obtained results with others from the literature and expand the discussion section. Adding a brief conclusion section could be also useful.
Author Response
Response:
We thank the referee for his review of our manuscript.
We have added the references in the places the referee has requested and defined the variables appearing in our manuscript. Error bars for the other figures, e.g. Fig 5B are of the same size as the markers used to plot the points.
The profiles in Fig. 4 of our manuscript clearly show that the surface migration is significantly suppressed, leading to lower migrant fraction at the surface, due to the gelation, both in the CGMD description (compare profiles in panels (a) and (b) of Fig. 4) and the mesoscale description (compare profiles in panels (c) and (d) of Fig. 4). The phase separation arrest, which originates in the bulk (compare the growth of the domain coarsening length scale presented in Figure 2) also affects the wetting behaviour when surface migration is coupled to the bulk phase separation. A discussion regarding the qualitative similarity of these profiles with those observed in the (surface directed spinodal decomposition) SDSD of mixtures of poly(ethylene-propylene)(PEP) and per-deuterated poly(ethylene-propylene) (d-PEP) has been added in the Results section of the modified manuscript.
We are therefore unsure why the referee is not convinced by our results, since she/he has not forwarded any arguments, physical, mathematical or computational to substantiate his belief.
We have clearly showed phase separational arrest in polymer gels which leads to reduced surface migration using coarse-grained molecular dynamics simulations. Using a coarse-grained mesoscale model with the bulk modulus being the only free parameter, we have obtained similar results.
To the best of our knowledge there are no studies that explore the effects of gelation on surface migration. We have cited some old work in context of phase separational arrest due to network formation discussed in context of alloys by Tanaka et al.
We have renamed the discussion section of the manuscript as conclusions as it seemed more appropriate.
Round 2
Reviewer 2 Report
In the response, the Authors indicate that “We agree with the referee's comments that calculations that are not supported by results obtained from tangible samples have very little significant value."
For this reason, I am surprised that the Authors submitted the manuscript without the experimental part. Without conducting an experiment that would validate the calculations, the work does not possess any significant scientific value and does not match the journal’s profile.
I suggest the Authors commence cooperation with a science team interested in conducting experimental research in aim to determine whether the described calculations can have practical application.
For this reason, I uphold my initial opinion and suggest a rejection of the submitted manuscript.
Reviewer 3 Report
After the modifications the article became understandable and readable. I have no additional comments to the Authors. I think that this manuscript can be published in the journal Polymers.
Reviewer 5 Report
The manuscript has been improved by the authors according to the reviewer suggestions. It is suitable for pubblication.